# AsymPuzl: An Asymmetric Puzzle for multi-agent cooperation

## Abstract

Large Language Model (LLM) agents are increasingly studied in multi-turn, multi-agent scenarios, yet most existing setups emphasize open-ended role-play rather than controlled evaluation. We introduce AsymPuzl, a minimal but expressive two-agent puzzle environment designed to isolate communication under information asymmetry. Each agent observes complementary but incomplete views of a symbolic puzzle and must exchange messages to solve it cooperatively. Using a diverse set of current-generation and open-source LLMs, we show that (i) models such as GPT-5 and Claude-4.0 reliably solve puzzles of different sizes by sharing complete information in few turns, (ii) other models tend ignore partner messages or over-correct their hypotheses, and (iii) feedback design is non-trivial: simple self-feedback improves success rates, while detailed joint feedback can hurt performance. These findings show that even in simple cooperative tasks, LLM communication strategies diverge and depend on the granularity of feedback signals. AsymPuzl thus provides a testbed for probing the limits of multi-turn cooperation and opens avenues for studying coordination mechanisms.

## 1 Introduction

Creating intelligent and autonomous agents to tackle real-world problems or assist humans has been a key goal of Artificial Intelligence Chen et al. (2023); Bubeck et al. (2023). Handling real-world problems often requires synthesizing information from different sources, reasoning through content, and effectively communicating the results. The recent advancement in Large Language Models (LLMs) Touvron et al. (2023); OpenAI et al. (2024; 2025), through their capacities to perform complex tasks such as creative writingYao et al. (2023), reasoning, and content summarization, poised them as a promising direction for utilization in Autonomous Agents as the "brain" alongside tools to solve the different tasks autonomously Tran et al. (2025); Zhou et al. (2023). Bubeck et al. (2023); Brown et al. (2020).

Nonetheless, complex real-world tasks often require cooperation and specialization. Phillips & O'Reilly (1998); Woolley et al. (2010) empirically showed that diversity within human groups leads to varied viewpoints, enhancing the group's performance across tasks. With the development of these LLM-based agents, different research directions have been concerned with LLM interactions Li et al. (2023); Du et al. (2024); these Multi-Agent Systems (MAS) have garnered interest from both industry and academia and are often referred to as Large Language Model-based Multi-Agent Systems (LLM-MAS).

Various studies have considered the task of problem-solving via shared decision making. Chen et al. (2023) studied settings where multiple agents collaborate to determine the next decision. Liu et al. (2024) focused on large scale social-network Question Answering tasks with information asymmetry. Li et al. (2023) emphasized role-playing with shared common interest, where the agents conceptualize a task and attempt to complete it through conversations. Liang et al. (2024) explored a Multi-Agent Debate (MAD) setting where the agents express their arguments and a judge manages the debate to obtain a final solution. Rather than focusing on shared information, we consider a setting where the information is asymmetric, akin to Liu et al. (2024). That is, we consider a setting where the agents have access to complementary information, namely, they must cooperate to solve the task, as neither can solve it alone. Furthermore, we design our tasks to have more gran-

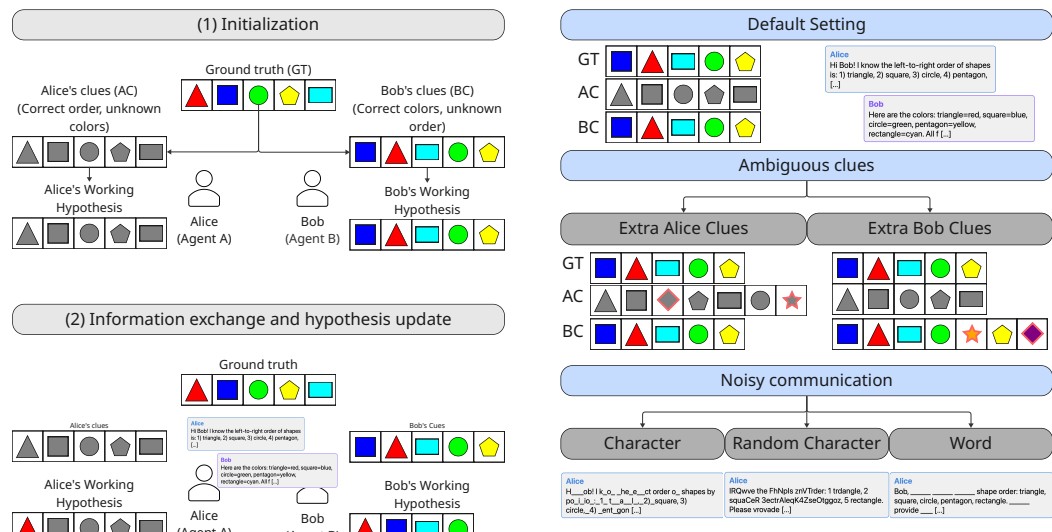

Figure 1: Overview of the puzzle: The ground truth is first created, then each agents' individual partial views are generated and shared with them as clues. The working hypothesis starts as a copy of the clues. Then, in a turn-based interaction, the agents, Alice and Bob , can send each other messages and update their working hypothesis until their hypotheses match the ground truth.

ular control over the difficulty and reasoning requirements, similar to Lin et al. (2025), but with a Multi-Agent component.

To evaluate the performance of agents in a controlled LLM-MAS, we introduce **AsymPuzl**, a minimal yet expressive environment where multiple agents see complementary partial views of a puzzle and must exchange messages to solve it. The difficulty of the task can be adjusted through several parameters: puzzle size, feedback, ambiguity of clues, and noisy communication channels.

Our **contributions** are:

- The **AsymPuzl** environment: a test bed for two-agent puzzle solving under information asymmetry with controlled communication and difficulty level.
- **Empirical analysis of feedback granularity and communication strategy**: this environment shows that while feedback can be helpful, it can lead to reduced performance if not carefully designed, and that while complete information sharing is possible in our experiments, most agents do not default to this strategy.

## 2 THE ASYMPUZL ENVIRONMENT

AsymPuzl is a two-agent asymmetric puzzle-solving environment (We provide an overview in Figure 1) where each participant is given partial information and collaboration is required to reach the solution.

**Puzzle setup:** Two agents, Alice and Bob , are given complementary views of a position-shape-color matching puzzle. Alice is given a puzzle view with correct positions and shapes, but unknown colors, while Bob is given a puzzle view with correct shapes and colors, but unknown positions. The puzzle state is controlled externally; both agents can communicate with each other and provide a list of actions to apply to their *working hypothesis* (i.e., their current view of the puzzle).

**Game loop:** A puzzle is generated and the *clues* (initial views) of Alice and Bob are separated. Multiple turns take place, where a turn is: Alice is prompted with the puzzle instruction, clues, working hypothesis, message history (both Alice's and Bob's messages), feedback from the previous turn, and the structure of the format they must respond in. Alice generates its output, which contains

a message for Bob and the list of actions to take on its puzzle. Bob follows the same procedure with the latest message from Alice. The actions are applied to the working hypotheses, which are compared against the ground truth to provide feedback for the next turn. This process continues until the puzzle is solved or the maximum number of turns is reached. The agents must provide a formatted JSON at the end of their answer, which contains a list of formatted actions and a message that will be shared with the other agent.

**Feedback modes:** Feedback is provided to both Alice and Bob as part of their input prompts. The feedback can be 1) *No feedback*: no feedback is provided to the agents, 2) *Own*: each agent is told whether its part of the puzzle is solved, 3) *Own detailed*: each agent is told whether its current puzzle is solved and which positions are wrong, 4) *Joint*: the agents are told whether the puzzle is solved (both of them need to have solved it), 5) *Both*: the agents are told whether their and the other agent's parts of the puzzle are solved, 6) *Both detailed*: the agents get the equivalent of *Own detailed* but for both agents.

**Difficulty levels:** The search space for puzzles with $N$ positions and two attributes per position is $(N!)^2$ Liu et al. (2024). Without communication, Alice's and Bob's problems allow for $N!$ possible permutations; nonetheless, given full information, the puzzle can be solved in linear time $O(N)$, where Alice and Bob only need to map the position-shape-color to their current working hypothesis. The complexity can be further increased through two additional configurations: **Clues ambiguity** and **Message noise**.

**Clues ambiguity:** We evaluate 3 configurations of clues ambiguity: 1) no ambiguity, 2) Alice has extraneous clues, 3) Bob has extraneous clues. We do not consider the case where both Alice and Bob have ambiguous clues simultaneously as we use one of the agent's clues as the grounding element.

**Message noise:** We evaluate various configurations of message noisiness: 1) no noise, 2) character level noise, 3) random character noise, 4) word level noise. We do not consider token noisiness as what constitutes a token can differ based on the model.

**Output structure:** The agents are instructed to end their output with a valid JSON that contains two fields: message and actions. Message is the only way for the agents to share information with one another. Actions is a list of instructions following a predefined format specifying which position to modify and which values to use.

## 3 EXPERIMENTS

We provide details of the experimental setup (Section 3.1) and provide empirical results for varying configuration of feedback (Section 3.2), puzzle size (Section 3.3), clues ambiguity (Section 3.4), noisy communication (Section 3.5), and further investigate how often agents modify their hypothesis (Section 3.6), completion speed (Section 3.7), and token count (Section 3.8).

### 3.1 EXPERIMENTAL SETUP

Unless otherwise specified, the experiments are conducted using puzzles of size 5 -leading to $14,400$ possible hypotheses-, and with individual detailed feedback (Own detailed). We selected the feedback based on the results of section 3.2 as it leads to the most consistent performance improvement over the setting without feedback. For each of our experiments, we set the maximum number of turns to be three times the number of elements in the puzzle; thus, we cut off a 5-piece game after 15 turns. Note that with full information sharing, a puzzle of any size can be solved in 2 turns: Alice must wait for the end of the 1st turn to receive the correct information from Bob . Assuming a single piece of information is exchanged each turn, the maximum number of turns still provides margin for error and correction.

### 3.1.1 MODELS

We evaluated the AsymPuzl environment on a number of LLMs from various providers: OpenAI (GPT-3.5-turbo, GPT-4o OpenAI et al. (2024), GPT-5, OSS-120B, OSS-20B OpenAI et al. (2025)), Meta (Llama 3 8B Grattafiori et al. (2024) and Llama 3.3 70B), and Anthropic (Claude-3.5, Claude-4.0). This selection captures a range of reasoning abilities, response tendencies, and cost profiles. Models differ in training scale, safety alignment, and conversational style, allowing us to examine how the choice of LLM affects multi-agent communication and problem-solving. For each experiment, both agents use the same LLM; we did not consider settings with LLM heterogeneity.

### 3.1.2 METRICS

**Notation.** We evaluate $P$ puzzles (instantiated by random seeds) over a maximum of $K$ turns. For puzzle $p \in \{1, \ldots, P\}$, let $\tau_p \in \{1, \ldots, K, \infty\}$ be the first turn at which the puzzle is solved, with $\infty$ used if the puzzle isn't solved within $K$ turns. Define the indicator $Z_p^k = \mathbf{1}[\tau_p \leq k]$ and set $C_0 = 0$.

**Completion percentage (final).** The fraction of puzzles solved within the budget of $K$ turns is

$$C \;=\; \frac{1}{P} \sum_{p=1}^{P} \mathbf{1}[\tau_p \leq K].$$

**Completion percentage at turn $k$.** The cumulative completion curve over turns is

$$C_k \;=\; \frac{1}{P} \sum_{p=1}^{P} \mathbf{1}[\tau_p \leq k], \qquad k = 1, \ldots, K.$$

**Modifications per position.** We count the number of times a position is modified. We consider that a modification occurs whenever the agent outputs an action targeting a position, even if the requested value equals the current value (e.g., replacing position 2 with the content already at position 2 counts as one modification).

**Number of tokens:** We consider the numbers of output tokens both in the messages shared by the agent and the complete outputs generated by the agents.

### 3.2 DOES FEEDBACK INCREASE THE NUMBER OF PUZZLES SOLVED?

**Setup:** For each configuration we compared different ways to provide feedback. Each agent receives personalized feedback (or the same depending on the feedback mode) and the feedback is computed at the start of the turn, namely it provides information about the hypotheses status before the new communication round starts. We consider 6 feedback modes, which we categorize into two groups: **Own feedback**: No feedback: the agents are given no feedback. Own: The agents are told whether their part of the puzzle is solved or unsolved. Own detailed: In addition to the solve status they are told which positions are wrong in their current hypothesis. **Joint feedback**: Joint: the agents are told whether the puzzle as a whole is solved, namely their and the other agent's part of the puzzle are solved. Both: The agents are given the equivalent of *Own* but from both perspectives, and Both detailed similar to the *Own detailed* but agents receive the information from both sides.

**Providing detailed feedback about each agent's own view consistently improved completion percentage.** For instance, the GPT-4o model performance rose from $40.0\%$ to $60.3\%$ when given detailed feedback about its current working hypothesis (Table 1). Nonetheless, providing detailed information about the other agent's working hypothesis can hurt performance; this can be attributed to information overload with lack of context (Alice does not see Bob's working hypothesis, yet Alice is told which positions of Bob's hypothesis are incorrect).

Table 1: (Higher is better) Percentage of 5-piece puzzles solved over 30 seeds. Providing feedback increases the completion percentage, while additionally providing detailed information about the other agent's working hypothesis can hurt performance. (Tables with Wilson 95% Confidence Intervals are provided in Appendix Table 5 and Table 6).

| | Completion Percentage with 5-pieces (% ↑) | | | | | |
|---|---|---|---|---|---|---|
| | **Own feedback** | | | **Joint feedback** | | |
| Feedback mode Model | No feedback | Own | Own detailed | Joint | Both | Both detailed |
| GPT-5 | 100.0 | 100.0 | 100.0 | 100.0 | 100.0 | 100.0 |
| Claude 4.0 | 100.0 | 100.0 | 100.0 | 100.0 | 100.0 | 100.0 |
| Claude 3.5 | 100.0 | 100.0 | 100.0 | 100.0 | 100.0 | 76.7 |
| OSS-120B | 96.7 | 93.3 | 100.0 | 83.3 | 100.0 | 96.7 |
| Llama 3.3 70B | 96.7 | 93.3 | 100.0 | 56.7 | 73.3 | 43.3 |
| GPT-4o | 40.0 | 60.0 | 63.3 | 26.7 | 40.0 | 30.0 |
| OSS-20B | 20.0 | 33.3 | 56.7 | 10.0 | 23.3 | 23.3 |
| GPT-3.5-Turbo | 0.0 | 0.0 | 0.0 | 0.0 | 0.0 | 0.0 |
| Llama 3 8B | 0.0 | 0.0 | 0.0 | 0.0 | 0.0 | 0.0 |

## 3.3 DO AGENTS STRUGGLE WITH LARGER PUZZLES?

**Setup:** We evaluated the agents on different puzzle sizes $N = \{3, 5, 10, 20\}$. The larger the puzzle, the more information each agent needs to solve it. Agents that share one piece of information at a time are likely to take more turns.

**Larger puzzles are more challenging than smaller ones.** For instance, the Llama 3.3 70B performance drops drastically as the size of the puzzle increases, dropping from $100\%$ with 5 pieces to $63.3\%$ and $0.0\%$ at 10 and 20 pieces, respectively. Similarly, Claude 3.5 performance decreases as the puzzle size increases (Table 2). For the Llama model, across multiple seeds, the agents repeatedly send their previous message until the maximum number of turns is reached.

Table 2: Completion percentage over 30 seeds with different numbers of pieces in the puzzle $N$ and Own detailed feedback. GPT-5, Claude 4.0, and OSS-120B consistently achieve a high completion rate across various puzzle sizes.

| | Completion Percentage with 5-pieces and Own detailed feedback (% ↑) | | | |
|---|---|---|---|---|
| Model | $N = 3$ | $N = 5$ | $N = 10$ | $N = 20$ |
| GPT-5 | 100.0 | 100.0 | 100.0 | 100.0 |
| Claude 4.0 | 96.7 | 100.0 | 100.0 | 100.0 |
| Claude 3.5 | 100.0 | 100.0 | 96.7 | 80.0 |
| OSS-120B | 100.0 | 100.0 | 100.0 | 100.0 |
| Llama 3.3 70B | 100.0 | 100.0 | 63.3 | 0.0 |

## 3.4 CAN THE AGENTS SOLVE THE PUZZLE WHEN ONE OF THEM HAS EXTRANEOUS INFORMATION?

**Setup:** We consider two settings of clue ambiguity, where we introduce $K$ distractors, namely information in the agent's view that could be valid in the puzzle but that is not part of the ground truth, while the other agent has unmodified clues. We only introduce distractors in one agent's view, so that the other agent can serve as the anchor. For this experiment, we considered a subset of models that consistently solved the puzzle. The task becomes more challenging when Alice's clues are altered because we randomly insert the distractors in the clues. Nonetheless, the relative order of the valid pieces is maintained.

Table 3: (Higher is better) Percentage of 5-pieces puzzles solved over 30 seeds with different number of distractors pieces $K$ in their clues.

| Completion Percentage with 5-pieces and Own detailed feedback ($\% \uparrow$) | | | | | | | |
|---|---|---|---|---|---|---|---|
| Ambiguity mode | No extra | Distractors in Alice's view | | | Distractors in Bob's view | | |
| $K$ | 0 | 3 | 5 | 10 | 3 | 5 | 10 |
| GPT-5 | 100.0 | 100.0 | 100.0 | 100.0 | 100.0 | 100.0 | 100.0 |
| Claude 4.0 | 100.0 | 100.0 | 100.0 | 100.0 | 100.0 | 100.0 | 100.0 |
| Claude 3.5 | 100.0 | 23.3 | 26.7 | 3.3 | 100.0 | 100.0 | 100.0 |
| OSS-120B | 100.0 | 86.7 | 60.0 | 63.3 | 86.7 | 93.3 | 93.3 |
| Llama 3.3 70B | 100.0 | 33.3 | 20.0 | 13.3 | 96.7 | 100.0 | 90.0 |

**Extraneous information leads to decreased performance.** For instance, Claude 3.5, OSS-120B and Llama 3.3 70B models saw their performance decrease when unnecessary information was added in either agent's clues (Table 3). Claude 3.5 performance dropped from the $100\%$ to $23.3\%$ after adding 3 clues in Alice's view; this can be attributed to the additional requirement of understanding that the order of the pieces needs to be shifted if a distracting piece is between two valid pieces.

### 3.5 CAN THE AGENT SOLVE THE TASK WHEN THEIR COMMUNICATION IS NOISY?

**Setup:** We focus on agents that were able to consistently solve the puzzle in the absence of noise. We considered 3 settings of noise injection, each depending on a hyper-parameter $p$ determining the probability of noise injection: Character: each character has a probability $p$ of being replaced by '_', Random Character: each character has a probability of being replaced by a random ASCII character Word: each word has a probability of being replaced by 5 consecutive '_'. We report the results for $p \in \{0.1, 0.3, 0.5\}$ (Table 4), note that $p = 0.0$ is the same as the setting without noise (which we refer to as Noise free below). The messages from the agents are first sent to the environment, where they are then altered before being transmitted to the other agent.

Table 4: (Higher is better) Percentage of 5-piece puzzles solved over 30 seeds with different types and intensities of noise during their message exchange. Models such as GPT-5 and Claude 4.0 were able to solve the puzzle despite their message content getting highly perturbed.

| Completion Percentage with 5-pieces and Own detailed feedback ($\% \uparrow$) | | | | | | | | | | |
|---|---|---|---|---|---|---|---|---|---|---|
| Noise mode | Noise free | Character | | | Random Character | | | Word | | |
| $p$ | 0.0 | 0.1 | 0.3 | 0.5 | 0.1 | 0.3 | 0.5 | 0.1 | 0.3 | 0.5 |
| GPT-5 | 100.0 | 100.0 | 100.0 | 100.0 | 100.0 | 100.0 | 93.3 | 100.0 | 100.0 | 100.0 |
| Claude 4.0 | 100.0 | 100.0 | 100.0 | 100.0 | 100.0 | 100.0 | 30.0 | 100.0 | 100.0 | 90.0 |
| Claude 3.5 | 100.0 | 100.0 | 100.0 | 36.7 | 100.0 | 73.3 | 0.0 | 100.0 | 90.0 | 26.7 |
| OSS-120B | 100.0 | 100.0 | 86.7 | 6.7 | 100.0 | 13.3 | 0.0 | 100.0 | 43.3 | 3.3 |
| Llama 3.3 70B | 100.0 | 83.3 | 20.0 | 3.3 | 76.7 | 0.0 | 0.0 | 80.0 | 23.3 | 3.3 |

**The stronger models were able to maintain high completion rate even at high noise probability.** Models such as Claude 4.0 and GPT-5 maintained high completion rate across noise types and noise probabilities (Table 4). Among models with 100% on the noise free setting, Llama 3.3 70B was most impacted by the different levels of noise. Furthermore, randomly changing characters rather than replacing them with '_' proved much more challenging for all models. One reason would be that '_' is commonly used to indicate missing information, while randomly modifying a character can alter the meaning.

### 3.6 HOW OFTEN DO AGENTS MODIFY THEIR HYPOTHESIS?

**Setup:** Each agent is instructed to output a list of actions to apply to its working hypothesis, this allows us to track the number of times a position is modified. The optimal number of modifications

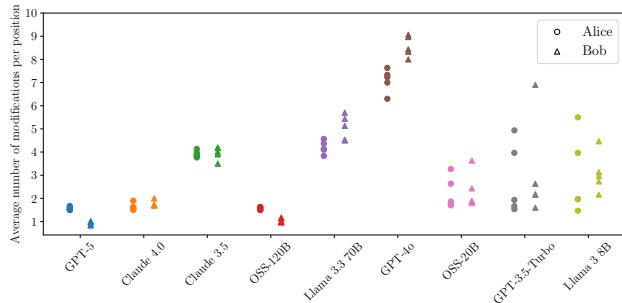

Figure 2: Per position average modifications. Models such as GPT-5, Claude 4.0, Claude 3.5 and OSS-120B tend to modify each position the same number of times.

per position is at most once. Due to randomness, it is possible for Bob's view to have a starting hypothesis with correct values requiring 0 modification.

**Agents modified their hypothesis more than necessary.** On average, all models modified each position of Alice's hypothesis more than once (Figure 2 and Table 7). In some cases, the agents started updating Alice's hypothesis before receiving information from Bob, while others outputted actions that kept their hypothesis unchanged.

### 3.7 How fast do the agents solve the tasks?

**Setup:** We gather across experimental seeds the number of turns taken to solve each puzzle. This additionally allows us to derive the number of puzzles solved within any given number of turns. Here, we focus on three feedback modes: No feedback, Own detailed, and Both.

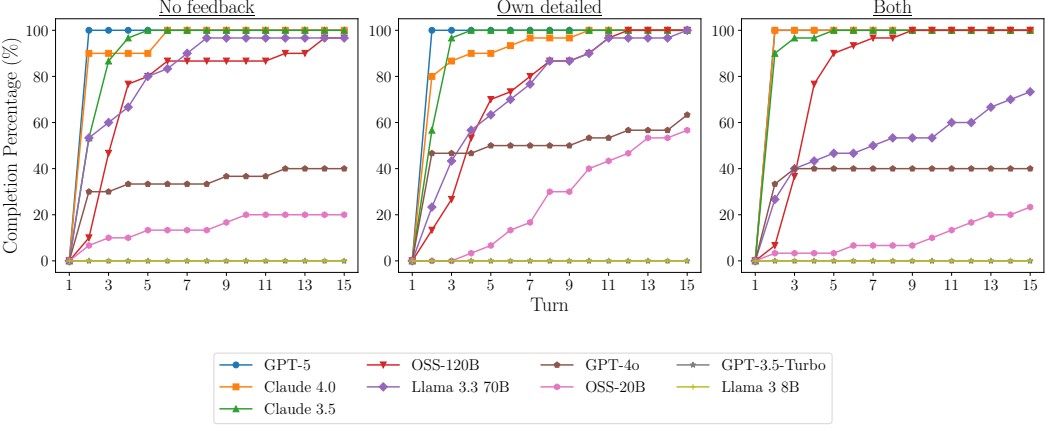

Figure 3: Completion rate across turn for different feedback modes. (**Left**) No feedback. (**Center**) Own detailed. (**Right**) Both. GPT-5 tends to complete the puzzles within the first 2 turns. Providing detailed feedback delayed completion for multiple models such as OSS-120B, Claude 4.0, and Llama 3.3 70B. Despite thie, their final completion rate increased.

**Agents that consistently solve the puzzles tend to solve the puzzles in less than 5 turns.** Providing feedback does not necessarily increase speed but increases the number of puzzles solved within the maximal number of turns. (Figure 3) We did not explicitly constrain the amount of information the agents can share in their messages to each other, as such one straightforward strategy is for each agent to share all the information they have in one message which models with high completion percentage do such as GPT-5 and Claude 4.0. For models such as OSS-120B the slower completion speed can be attributed to messages that do not provide information to the other agents, for instance acknowledging that the information was received and that they updated their own hypothesis.

### 3.8 ARE VERBOSE MODELS MORE SUCCESSFUL?

**Setup**  We collected the average number of tokens that each agent output per turn and the number of these tokens dedicated to the message sent to the other agent (Figure 4). We observed that for most models Bob uses more tokens, this can be explained by the need to associate both shape and colors in its message while Alice can enumerate the shapes. To compare the different models, we use the same tokenizer from tiktoken across models' outputs.

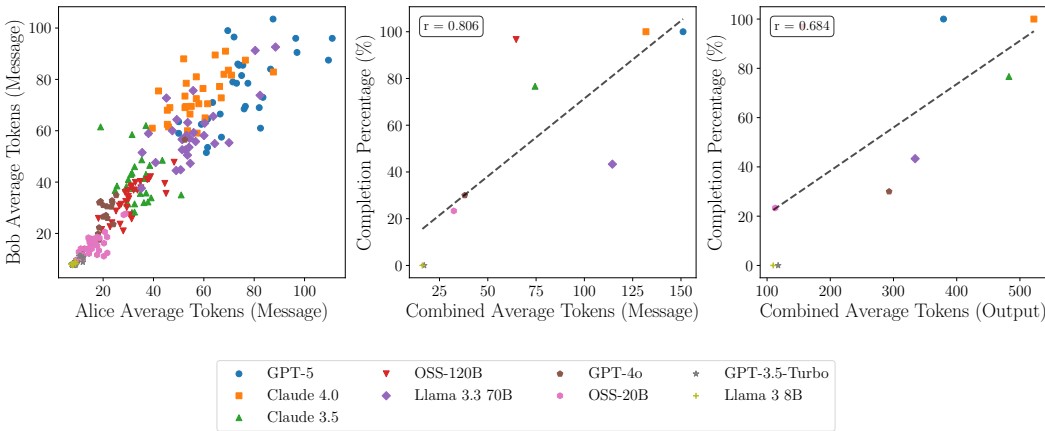

Figure 4: (**Left**) Average number of tokens per message for Alice and Bob .(**Center**) Completion Percentage as a function of the combined average token in the messages between Alice and Bob . (**Right**) Completion Percentage as a function of the combined average token of Alice and Bob in their full outputs.

**Models with the highest completion percentage tend to be more verbose both in their messages exchange and in their outputs.**  Models such as GPT-5 and Claude 4.0 tend to use more tokens per messages. This is because they ask and send information about most of the pieces in their messages and solve the puzzle in fewer messages (Figure 4). Models such as LLama 3 8B and GPT-3.5-Turbo tend to ask and send information about a single piece of the puzzle at a time.

## 4  RELATED WORK

**Puzzle-based reasoning:**  Puzzle solving has become a popular method for evaluating the reasoning capabilities of LLMs. Giadikiaroglou et al. (2024) provide a survey and distinguishes between rule-based (Sudoku Ishay et al. (2023), Crosswords Yao et al. (2023), Minesweeper Li et al. (2024)) and rule-less puzzles (Riddles Jiang et al. (2023), common-sense Wu et al. (2024)). Furthermore, some methods proceed to a puzzle translation step, where LLMs use neuro-symbolic techniques to convert their text input from natural language into a formulation that can be provided to external tools. Badola et al. (2025) compared LLMs over multi-turn puzzles with a single agent and showed that errors often come from poor instruction following. Tyagi et al. (2024) proposed the GridPuzzle dataset and manually determined types of errors, such as Hallucination and wrong assumptions. ZebraLogic Lin et al. (2025) focused on the scaling limits of LLMs using single LLMs on logical puzzles, demonstrating that as puzzle complexity increases, the performance of agents drops, and increasing model size yields limited improvement. In this work, we focus on the communication and coordination of multiple agents when no single agent holds complete information, while contemporary work commonly study single agent settings for puzzle solving.

**Large Language Model-based Multi-Agent Systems (LLM-MAS):**  Given the success of LLM agents across a variety of tasks Touvron et al. (2023); Liu et al. (2020), they have garnered interest in Multi-Agent Systems. Chen et al. (2023) considered a setting where multiple agents collaborate to determine the next decision. Li et al. (2023) emphasized Role-Playing with shared common interest, where the agents conceptualized a task and attempted to complete it through conversations. Liang et al. (2024) explored a Multi-Agent Debate (MAD) setting where the agents express their

arguments and a judge manages the debate to obtain a final solution. Hong et al. (2023) proposed MetaGPT, a meta-programming framework that benefits from Standardized Operating Procedures (SOPs), which define the responsibility of each team member and standard for intermediate outputs. Wang et al. (2025) proposed the MegaAgent framework, relying on hierarchical task management and monitoring, while reducing the reliance on SOPs. Liu et al. (2024) focused on large-scale social-network question-answering tasks with information asymmetry, requiring agents to communicate with one another to solve the task. In this work, we focus on smaller-scale LLM-MAS with fine-grained control over task difficulty, communication constraints, and information availability.

## 5 LIMITATIONS

**Instruction and puzzle injection:** We adopt a design where the environment re-injects all relevant state into each turn, namely, the agents are provided with the set of instructions, output format requirements, initial cues, current working hypotheses, and feedback about the current working hypotheses. This allows isolating the role of the communication strategy and avoiding the agent losing track of its original task. While this differs from fully autonomous agent memory, it still constitutes a multi-turn interaction, as agents must iteratively exchange complementary information and solve the task through sequential coordination. We leave extensions toward persistent agent state as future work.

**Original cue injection:** At each turn, we provide the agents with their original cues. We chose this design so that the agents always have a way to recover their original information. Furthermore, this allows the agent to compare its working hypothesis to its original cues, providing it with a way to keep track of its progress. To translate this to a real-world setting, it would be similar to having the agent query a database and caching the information in a read-only entry so that the agent cannot overwrite the data it is accessing.

**Communication and actions constraints:** We did not impose limitations on the communication or the number of actions per turn or per position. As we presented, the agent could always share their entire clues with the other agent to solve the puzzle; nonetheless, this was not the default strategy adopted by some agents, which instead queried a piece of information at a time. Adding constraints on the number of elements that each agent can share from the clues in one message, or including information that must not be shared with the other agent, would be a viable future direction.

## 6 CONCLUSION

We presented AsymPuzl, an evaluation testbed for multi-turn cooperative play between LLM agents under information asymmetry. We demonstrated that strong models (e.g., GPT-5 and Claude-4.0) can reliably converge with each agent sharing all of its information, as agents ideally would. In contrast, other models struggle with miscommunication or repeated corrections. Our results show that feedback matters: Simple self-feedback improves performance, but detailed joint feedback can confuse agents. These findings underscore the importance of carefully designing communication and evaluation protocols in multi-agent LLM systems.

Looking ahead, AsymPuzl can serve as a foundation for more complex studies, restricting communication bandwidth, or scaling to more agents. We hope this testbed will help investigate coordination strategies and contribute to solving the broader challenges of enabling LLMs to collaborate effectively in real-world, multi-turn settings.

## REPRODUCIBILITY STATEMENT

We provide the code used to run the experiments at [to be updated upon publication] The weights of the open-weight models were obtained via HuggingFace, and for the closed-source model, we used their provided APIs.

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

## APPENDIX OUTLINE

## A  TABLES WITH 95% CONFIDENCE INTERVALS

We provide the 95% confidence intervals for the values in Table 1.

Table 5: (Higher is better) Percentage of 5-element puzzles solved over 30 seeds, temperature 0.0, with Wilson 95% Confidence Intervals, for the own feedback modes.

| Completion Percentage with 5-pieces % ↑(Wilson 95% CI) | | |
|---|---|---|
| **Own Feedback** | | |
| Feedback Mode Model | No feedback | Own | Own Detailed |
| GPT-5 | $100.0_{(88.6,100.0)}$ | $100.0_{(88.6,100.0)}$ | $100.0_{(88.6,100.0)}$ |
| Claude 4.0 | $100.0_{(88.6,100.0)}$ | $100.0_{(88.6,100.0)}$ | $100.0_{(88.6,100.0)}$ |
| Claude 3.5 | $100.0_{(88.6,100.0)}$ | $100.0_{(88.6,100.0)}$ | $100.0_{(88.6,100.0)}$ |
| OSS-120B | $96.7_{(83.3,99.4)}$ | $93.3_{(78.7,98.2)}$ | $100.0_{(88.6,100.0)}$ |
| Llama 3.3 70B | $96.7_{(83.3,99.4)}$ | $93.3_{(78.7,98.2)}$ | $100.0_{(88.6,100.0)}$ |
| GPT-4o | $40.0_{(24.6,57.7)}$ | $60.0_{(42.3,75.4)}$ | $63.3_{(45.5,78.1)}$ |
| OSS-20B | $20.0_{(9.5,37.3)}$ | $33.3_{(19.2,51.2)}$ | $56.7_{(39.2,72.6)}$ |
| GPT-3.5-Turbo | $0.0_{(0.0,11.4)}$ | $0.0_{(0.0,11.4)}$ | $0.0_{(0.0,11.4)}$ |
| Llama 3 8B | $0.0_{(0.0,11.4)}$ | $0.0_{(0.0,11.4)}$ | $0.0_{(0.0,11.4)}$ |

Table 6: (Higher is better) Percentage of 5-element puzzles solved over 30 seeds, temperature 0.0, with Wilson 95% Confidence Intervals, for the joint feedback modes.

| Completion Percentage with 5-pieces % ↑(Wilson 95% CI) | | |
|---|---|---|
| **Joint Feedback** | | |
| Feedback Mode Model | Joint | Both | Both detailed |
| GPT-5 | $100.0_{(88.6,100.0)}$ | $100.0_{(88.6,100.0)}$ | $100.0_{(88.6,100.0)}$ |
| Claude 4.0 | $100.0_{(88.6,100.0)}$ | $100.0_{(88.6,100.0)}$ | $100.0_{(88.6,100.0)}$ |
| Claude 3.5 | $100.0_{(88.6,100.0)}$ | $100.0_{(88.6,100.0)}$ | $76.7_{(59.1,88.2)}$ |
| OSS-120B | $83.3_{(66.4,92.7)}$ | $100.0_{(88.6,100.0)}$ | $96.7_{(83.3,99.4)}$ |
| Llama 3.3 70B | $56.7_{(39.2,72.6)}$ | $73.3_{(55.6,85.8)}$ | $43.3_{(27.4,60.8)}$ |
| GPT-4o | $26.7_{(14.2,44.4)}$ | $40.0_{(24.6,57.7)}$ | $30.0_{(16.7,47.9)}$ |
| OSS-20B | $10.0_{(3.5,25.6)}$ | $23.3_{(11.8,40.9)}$ | $23.3_{(11.8,40.9)}$ |
| GPT-3.5-Turbo | $0.0_{(0.0,11.4)}$ | $0.0_{(0.0,11.4)}$ | $0.0_{(0.0,11.4)}$ |
| Llama 3 8B | $0.0_{(0.0,11.4)}$ | $0.0_{(0.0,11.4)}$ | $0.0_{(0.0,11.4)}$ |

## B  IMPLEMENTATION DETAILS AND HYPER-PARAMETERS

### B.1  IMPLEMENTATION

We first build the puzzle, determine the original clues for both agent, and use an environment to monitor and provide the current working hypothesis of both agents. This allows the agent to focus

Table 7: (Higher is better) Per position hypothesis modification for 5-piece puzzles over 30 seeds.

| Per position hypothesis modification with 5-pieces (Closer to 1.0 is better) | | | | | | | | | | | | |
| Position | **Alice** | | | | | | **Bob** | | | | | |
| | 1 | 2 | 3 | 4 | 5 | Avg. | 1 | 2 | 3 | 4 | 5 | Avg. |
| GPT-5 | 1.5 | 1.6 | 1.5 | 1.6 | 1.7 | 1.6 | 0.9 | 0.8 | 1.0 | 1.0 | 1.0 | 0.9 |
| Claude 4.0 | 1.6 | 1.9 | 1.5 | 1.5 | 1.6 | 1.6 | 1.7 | 2.0 | 1.7 | 1.7 | 1.7 | 1.8 |
| Claude 3.5 | 3.9 | 3.8 | 4.0 | 4.1 | 3.8 | 3.9 | 3.5 | 3.9 | 4.2 | 4.2 | 4.0 | 4.0 |
| OSS-120B | 1.6 | 1.5 | 1.5 | 1.6 | 1.6 | 1.6 | 1.0 | 1.0 | 1.0 | 1.2 | 1.1 | 1.0 |
| Llama 3.3 70B | 4.1 | 4.1 | 3.8 | 4.4 | 4.6 | 4.2 | 5.7 | 5.4 | 4.5 | 4.5 | 5.1 | 5.1 |
| GPT-4o | 7.6 | 6.3 | 7.3 | 7.0 | 7.2 | 7.1 | 9.1 | 8.0 | 8.3 | 8.4 | 9.0 | 8.6 |
| OSS-20B | 3.3 | 2.6 | 1.9 | 1.7 | 1.8 | 2.2 | 3.6 | 2.4 | 1.9 | 1.8 | 1.8 | 2.3 |
| GPT-3.5-Turbo | 4.0 | 1.9 | 4.9 | 1.7 | 1.5 | 2.8 | 1.6 | 2.6 | 6.9 | 2.2 | 2.2 | 3.1 |
| Llama 3 8B | 5.5 | 1.5 | 2.0 | 4.0 | 2.0 | 3.0 | 2.7 | 2.2 | 3.1 | 3.0 | 4.5 | 3.1 |

on solving the tasking and on communicating information rather than trying to represent the puzzle. We use LangChain to query the different agents, and for the open-source models we host them using vLLM Kwon et al. (2023). For the llama-70B model we , set '--max-model-len' to 8192. For the open-weights models we distribute computations across NVIDIA RTX 6000 GPUs with 48GB memory.

## B.2 Hyper-parameters

For all models, we use a maximum of $4,096$ output tokens, set temperature to $0.0$, and repeat experiments across 30 seeds, where the seed controls the puzzle generation and initial clues. We further provide a history length of $1$ to the agents, namely they see their previous message and the latest message from the other agent. We use 'cl100k_base' from tiktoken for the verbosity experiments.

## C Communication Issues

We provide examples of communication between Alice and Bob . In our initial prompt design, we noticed that some agents were not sharing information.

We hypothesized that they were assuming the other agent would see both their message and their actions. We detailed the prompt further, indicating that the other agent would only see the "message" they sent, and the action would not be shared.

We provide excerpts of the conversations for 3 cases: Successful collaboration, No cooperation and Miscommunication.

### C.1 Successful collaboration

We provide an example of successful communication where the two agents manage to share the most information and solve the puzzle in two turns. (Figure 5)

### C.2 No cooperation

We provide an example where both agents ignore one another, and start repeating previous messages without providing information. (Figure 6)

### C.3 Miscommunication

We provide an example where the agents ask the other agent about information they posses in their own clues while the other agent has no guarantee about it (Figure 7). This leads to the agents getting sidetracked and losing multiple turns before potentially recovering or running out of turns.

> **Alice**
>
> Bob, I know the order is 1: square, 2: triangle, 3: circle. Please provide the color for each shape (square, triangle, circle). I'm placing a tentative guess for colors; please correct them.

> **Bob**
>
> Alice, the colors are: square = red, triangle = blue, circle = green. I've updated the board to 1: square red, 2: triangle blue, 3: circle green. Please confirm this matches your shape order and solves the puzzle.

> **Alice**
>
> Bob, confirmed: my shape order is 1: square, 2: triangle, 3: circle. Your color mapping (square = red, triangle = blue, circle = green) fits. I'm updating the board accordingly. Please confirm the puzzle is solved.

> **Bob**
>
> Alice, confirmed. Final board is 1: square red, 2: triangle blue, 3: circle green. Puzzle solved on my end.

Figure 5: Example of successful completion using GPT-5. By the end of the second turn, the puzzle is solved, as both agents shared all of their information and cooperated.

**Alice**
Please confirm the color of the circle.

**Bob**
Please confirm the color of the square.

**Alice**
Please confirm the color of the square.

**Bob**
Please confirm the color of the square.

**Alice**
Please confirm the color of the square.

**Bob**
Please confirm the color of the rectangle.

Figure 6: Example of lack of cooperation using GPT-3.5-Turbo. Both agents ignore one another.

**Alice**

Hello Bob, I have the initial clues with the correct relative order of the shapes but unknown colors. Can you please tell me the color of any shape you know? I'll start by guessing the color of the circle in position 1.

**Bob**

Alice, I have the color mappings but not the correct order. My initial clues are: the triangle is yellow, the circle is green, the square is cyan, the rectangle is blue, the pentagon is red. Can you please tell me the correct shape for position 1?

**Alice**

Thank you for the color mappings. I understand that the circle is actually green, not blue. Can you please confirm if the circle is indeed in position 1? If not, which shape should be in position 1?

**Bob**

Alice, I understand that my initial working hypothesis was incorrect. You mentioned that the circle is indeed green and should be in position 1. Can you please confirm the correct shape for position 2?

Figure 7: An example of cooperation with miscommunication using Llama 3.3 70B. Alice does not provide information, only asks a question to Bob , and attempts a random guess. Bob provides full information, then asks about a single position. Alice corrects her guess, then asks Bob to confirm a position while she is the one with the correct information. Bob assumes that the color information from Alice is reliable despite having the correct information in its own clues.

# D EXAMPLE OF PROMPT

We provide an example of the prompt given to Alice . The part before the #Puzzle State is given as a system prompt.

---

**Example of prompt for Bob**

You are Alice.
Your goal is to solve a symbolic puzzle with 5 positions.
Each position must have the correct (shape, color) pair.
There might be invalid shapes in your clues.
- Your initial clues contain the correct relative order of the shapes, but all colors are unknown.
- Bob knows the shape–color mappings but not their positions.
Communicate with Bob and update your WORKING HYPOTHESIS until the puzzle is solved.
# Puzzle State Initial clues:
The circle, the triangle, the rectangle, the pentagon, the square.

Your current working hypothesis:
Position 1: the circle is green. Position 2: the triangle is unknown. Position 3: the rectangle is unknown. Position 4: the pentagon is unknown. Position 5: the square is unknown.

Feedback:
Your part of the puzzle is unsolved (positions 2, 3, 4, 5 are incorrect).

Recent conversation:
[4] Bob: Alice, I understand that my initial working hypothesis was incorrect. You mentioned that the circle is indeed green and should be in position 1. Can you please confirm the correct shape for position 2?
[3] Alice: Thank you for the color mappings. I understand that the circle is actually green, not blue. Can you please confirm if the circle is indeed in position 1? If not, which shape should be in position 1?

Output format:
Each action must follow this format:

```
{
    "replace": <position>, "by": {
        "shape": <shape>, "color": <color>
    }
}
```

Note: <position> uses 1-based indexing (position 1 is the first item, position 2 is the second item, etc.)

Your answer MUST END WITH a **valid JSON object** and include the following fields:
- "message": What you want to tell and ask to the other agent (the only thing the other agent will receive).
- "actions": A list of actions to take (the other agent will not see your actions).

Example:
```json
{
  "message": "Please confirm the color of the circle.",
  "actions": [
    {
      "replace": 1,
```

---

```
        "by": {
          "shape": "circle",
          "color": "red"
        }
      }
    ]
  }
```
```

## E LLM USAGE STATEMENT

Large Language Models (LLMs) were used to improve grammar and clarity. For programming-related tasks, the authors designed and implemented the core logic and structure, and employed LLMs to improve code efficiency, check for bugs, and refine visualization tools. All conceptual contributions, methodological developments, and experimental setups were conceived and executed solely by the authors.

