# OpenReview forum: "AsymPuzl: An asymmetric puzzle for multi-agent cooperation"
_ICLR.cc/2026/Conference — ICLR 2026 Conference Withdrawn Submission_

### Official Review · Reviewer_v6Uz · 2025-10-30

**Soundness:** 3
**Presentation:** 3
**Contribution:** 2
**Rating:** 2
**Confidence:** 4

**Summary:**

The paper introducing AsymPuzl, a two-agent puzzle environment designed to evaluate multi-agent cooperation and communication strategies among LLM agents. In this setup, two agents, hold complementary, partial views of a symbolic puzzle and must exchange messages to solve it collaboratively. The study explores how variables like feedback granularity, puzzle size, information ambiguity, and communication noise affect LLM performance across various models, including advanced versions like GPT-5 and Claude-4.0. Key findings indicate that while simple self-feedback generally improves success rates, highly detailed joint feedback can actually hinder performance, revealing that even in simple tasks, LLM communication strategies diverge significantly.

**Strengths:**

- The paper is well-structured and clearly written, making it accessible to readers with varying levels of familiarity with multi-agent systems and LLM research. The paper provide sufficient background context and clearly articulate their research questions and methodology.
- The AsymPuzl game design is elegant and intuitive, providing a straightforward yet effective framework for measuring cooperation and communication between two LLM agents. The puzzle's simplicity allows for controlled experimentation while still capturing essential aspects of collaborative problem-solving, such as information asymmetry and the need for effective communication protocols.
- The proposed framework serves as a testbed for evaluating multi-turn cooperation and provides opportunities for studying coordination mechanisms

**Weaknesses:**

- From my perspective, the proposed paper does not fit well within the scope of ICLR as it primarily focuses on evaluating the capabilities of existing LLMs through prompting strategies rather than introducing novel methodologies, architectures, or learning algorithms. While the AsymPuzl environment itself is a useful contribution, the paper's core findings are largely empirical observations about how different LLMs perform under various communication conditions. The work would benefit from proposing new training techniques, communication protocols, or theoretical frameworks that could advance the field of multi-agent learning beyond benchmarking existing models.
- The systems are only tested for a limited number of N values, and the results do not appear to converge in several cases. For example, in Table 2, the performance metrics show considerable variation, but the experiments do not extend to sufficiently large values of N to determine whether performance trends stabilize. Additional experiments with larger N would help establish whether the observed patterns are fundamental limitations of current LLMs or artifacts of the specific scale tested.
- The system is tested exclusively with pure LLMs without incorporating additional tools, external memory, or agent-based architectures. This limitation is significant because real-world multi-agent systems often leverage specialized tools, retrieval mechanisms, or structured reasoning frameworks to enhance performance. Testing the AsymPuzl environment with augmented agents would provide valuable insights into whether the observed communication challenges are inherent to language-based coordination or can be mitigated through tool use.

**Questions:**

What are relevant use cases where the evaluated puzzle soving capabilites would be relevant?

---

### Official Review · Reviewer_y6Vg · 2025-10-30

**Soundness:** 3
**Presentation:** 3
**Contribution:** 2
**Rating:** 2
**Confidence:** 4

**Summary:**

The paper proposes a puzzle for two agents. The puzzle is about finding the correct order and colors of different shapes. One agent is given hints about the correct order. The other is given hints about the correct colors. The paper then investigates how the two agents share information about their hints to solve the puzzle. It computes the success rate, number of turns before success, etc.

**Strengths:**

- The game has a clear ground truth that allows computing baselines. It can be relatively increased in difficulty by adding more objects. It investigates a setup with distractors, such as including objects in one of the hints that are not in the ground truth.

**Weaknesses:**

- Larger models such as GPT-5 consistently perform near-perfectly in most of the experiments. Since the paper mentions that one of its contributions is providing a testbed for agent coordination, this makes this testbed less likely to be useful for studying advanced models.

- The puzzle does not necessarily involve sophisticated levels of multi-turn or strategic coordination and an inherent need for having more than one agent. As the paper mentioned repeatedly, the puzzle can be solved by exchanging the complete view/hints of each agent in one turn. There is nothing that strictly enforces that some parts of the hints are private, can't be shared, etc. There is also no need for repeated turns. These factors make the puzzle quite simplistic and not complex enough for any real-world scenario or for advanced models. The paper also mentions that the setup requires agents to operate under partial information, but this is not the case.

- The paper introduces factors that would increase the complexity, such as inserting random characters in the communication between agents. But these factors are artificial (in the sense that they don't relate to the complexity of the puzzle itself). Also, this does not measure the coordination skills or capabilities exclusively, but rather the understanding of models and the denoising of the natural language. Even for small models, random characters can completely break the tokenization; the degradation (or the increase in the number of turns) is not directly related to a lack of coordination.

- There are multiple related works on this area of LLM multi-agent coordination (see, for example, https://arxiv.org/pdf/2310.03903 NAACL 2025) that the paper does not compare to. The necessity/novelty of this particular testbed is not clearly explained.

**Questions:**

- Are agents told that the communication will be adjusted? It is imaginable that in this case, the agents may try to embed the information multiple times.

- Similarly, are agents told that they should optimize the number of turns or tokens?

---

### Official Review · Reviewer_BL4G · 2025-10-31

**Soundness:** 2
**Presentation:** 2
**Contribution:** 1
**Rating:** 2
**Confidence:** 5

**Summary:**

This paper presents AsymPuzl, a controlled evaluation environment for studying multi-turn cooperation between large language model agents under information asymmetry.
Motivated by the observation that most existing multi-agent setups focus on open-ended role-play rather than measurable cooperation. the work aims to systematically examine how communication strategies and feedback mechanisms affect problem-solving when agents hold complementary but incomplete information.

The core method is a two-agent symbolic puzzle environment where each agent receives asymmetric clues and must iteratively communicate and update its hypothesis to reconstruct the full solution. The framework allows precise control over puzzle size, feedback type, clue ambiguity, and communication noise, enabling a structured analysis of coordination behaviors. The authors evaluate a diverse set of models and report how feedback granularity, message verbosity, and communication robustness influence overall success.

Results show that strong models such as GPT-5 and Claude-4.0 can reliably solve puzzles through efficient information sharing in few turns, while weaker models struggle with under-communication or over-correction. Simple self-feedback improves performance, whereas overly detailed joint feedback reduces it.
The study concludes that in multi-agent cooperation, communication strategy is as critical as model capability. AsymPuzl serves as a foundational testbed for future work on coordination mechanisms, communication efficiency, and scalable multi-agent systems.

**Strengths:**

1. The puzzle task is well-designed, simple, and interpretable. It provides a quantifiable and verifiable objective, making it suitable for controlled evaluation of cooperative reasoning.

2. The environment allows precise manipulation of task difficulty and reasoning requirements through parameters such as puzzle size, feedback type, clue ambiguity, and communication noise. This enables fine-grained analysis of agent cooperation.

3. The authors conduct extensive evaluation across a diverse set of mainstream LLMs, offering a broad and informative comparison that strengthens the empirical validity of the results.

**Weaknesses:**

1.	The paper lacks originality. Prior studies have already explored multi-agent cooperation under asymmetric information with a wide range of interaction mechanisms, and this work does not present any clear conceptual or methodological innovation.

2.	The main contribution lies in the AsymPuzl environment for studying cooperation under asymmetric information and controlled communication. However, most findings, such as “larger puzzles are more challenging,” offer little new knowledge or insight to the community.

3.	The claim that detailed feedback about another agent’s hypothesis can reduce performance is not convincing. As the authors acknowledge, this effect may stem from context fragmentation and information overload. A more complete experimental setup would strengthen the claim; as it stands, the setting feels underdeveloped and unable to robustly support any conclusion.

4.	Most results are purely descriptive and lack deeper analysis, for example in lines 341 to 344.

5.	The AsymPuzl environment is overly simplified, which limits the generalizability of its findings to real-world multi-agent systems.

6.	Overall, the paper makes little substantive contribution. While AsymPuzl may serve as a tool-level addition, the asymmetric-information setup is not novel, and the analyses are too shallow to constitute meaningful scientific progress.

**Questions:**

-

---

### Official Review · Reviewer_Wo24 · 2025-11-01

**Soundness:** 2
**Presentation:** 2
**Contribution:** 3
**Rating:** 4
**Confidence:** 3

**Summary:**

AsymPuzl introduces a two-agent asymmetric puzzle to study cooperation between LLMs. Each agent has partial information (order / colors) and must communicate over turns to reconstruct the full solution. Experiments with various LLMs shows that strong models share information efficiently and solve large puzzles, while weaker ones miscommunicate or over-revise (spend too much turns).

**Strengths:**

1. Asymmetric puzzle solving with communication is an interesting topic for LLMs and would be valuable for future multi-agent system studies. I appreciate the reseach topic.
2. There are experiments across major open and closed LLMs. Also, there are various settings proposed, such as ambiguous / extra clues, significantly enriching the proposed benchmark.

**Weaknesses:**

1. The design of AsymPuzl is not good enough. It is not a setting that naturally requires multi-round communication. In the base configuration, if both agents simply share all their information during the first communication round, the problem essentially degenerates into a single-agent puzzle-solving task. I believe the environment should introduce a more clever form of partial observability so that communication emerges naturally. Currently, adding noise to the communication channel feels like a separate topic — more about studying noisy or robust communication rather than asymmetric puzzle solving itself.

2. The difficulty level also needs to be increased. Currently, GPT-5, Claude-4.0/3.5, and GPT-OSS-120B can already solve this benchmark quite well, which reduces its potential research value.

3. In Figure 1, some of the text is too small to read clearly.

**Questions:**

1. Can you also include a template-based communication baseline? For example, a simple strategy where agents share all available information in the first round.
2. What would happen if the agents were heterogeneous — for example, one GPT-5 and one GPT-3.5?
3. Could you provide more communication examples? They don’t need to be shown in the same format as in the appendix — a more compact table or text-box layout with multiple examples would be sufficient.

---

### Note · Authors · 2025-11-18

I have read and agree with the venue's withdrawal policy on behalf of myself and my co-authors.